# Effect of Dendrite Fraction on the $M_{23}C_6$ Precipitation Behavior and the Mechanical Properties of High Cr White Irons

**Jun-Seok Oh [1], Young-Gy Song [1], Baig-Gyu Choi [2], Chalothorn Bhamornsut [3], Rujeeporn Nakkuntod [3], Chang-Yong Jo [4,*] and Je-Hyun Lee [1,\*]**

[1] Department of Advanced Materials, Changwon National University, Changwon 51140, Korea; jsoh5@changwon.ac.kr (J.-S.O.); syg222@changwon.ac.kr (Y.-G.S.)

[2] High Temperature Materials Department, Korea Institute of Materials Science, Changwon 51508, Korea; choibg@kims.re.kr

[3] Thailand Institute of Scientific and Technological Research, Thani 12120, Thailand; chalothorn@tistr.or.th (C.B.); rujeeporn@tistr.or.th (R.N.)

[4] Department of Advanced Materials, Dongeui University, Busan 47340, Korea

\* Correspondence: cyj20@deu.ac.kr (C.-Y.J.); ljh@changwon.ac.kr (J.-H.L.); Tel.: +82-51-890-2285 (C.-Y.J.); +82-55-213-3695 (J.-H.L.)

**Abstract:** High Cr white irons with various fractions of primary dendrite have been prepared through the modification of their chemical composition. Increasing C and Cr contents decreased the primary dendrite fraction. Eutectic solidification occurred with the phase fraction ratio of austenite: $M_7C_3$ = 2.76:1. The measured primary dendrite fractions were similar to the calculated results. ThermoCalc calculation successfully predicted fractions of $M_7C_3$, austenite, and $M_{23}C_6$. Conventional heat treatment at high temperature caused a destabilization of austenite, releasing it's solute elements to form $M_{23}C_6$ carbide. Precipitation of $M_{23}C_6$ during destabilization preferentially occurred within primary (austenite) dendrite, however, the precipitation scarcely occurred within austenite in eutectic phase. Thus, $M_{23}C_6$ precipitation by destabilization was relatively easy in alloys with a high fraction of primary dendrite.

**Keywords:** high chromium white irons; dendrite volume fraction; carbon equivalent; eutectic carbide; ThermoCalc; hardness; wear resistance; directional solidification and quenching

## 1. Introduction

The mechanical properties of high Cr white cast irons are governed by not only their matrix structure but also by existing carbides. Like solidification of hypoeutectic gray irons, the hypoeutectic high Cr white irons solidify primarily with formation of austenite dendrites followed by austenite and carbide eutectic reaction [1]. The carbides in high Cr white irons are very hard and wear resistant but very brittle [2]. Wear resistance could be improved by increasing the carbide amount. The types of existing carbides in high Cr irons are $M_7C_3$, $M_3C$, and $M_{23}C_6$. Generally, the eutectic reaction generates $M_7C_3$ in high Cr white iron. $M_7C_3$ carbides contribute to the improvement of wear resistance of the alloy, however those primarily precipitated from the melt ahead of eutectic reaction are known to be quite deleterious to impact toughness and should be avoided [2]. Thus, the hypereutectic composition of the alloys is not desirable for engineering applications. In the hypoeutectic alloys, increasing carbide fraction reduces toughness. However, the higher fraction of austenite (or dendrite) improves toughness but is detrimental to hardness of the alloy due to the reduction of the $M_7C_3$ fraction. The impact property of the alloy is dependent upon the microstructural balance between the fraction of austenite (will transform to martensite) matrix and carbide.

It is known that the fractions of austenite and $M_7C_3$ carbide in hypoeutectic high Cr white iron are keen to C content and Cr content [2]. Decreasing C content and Cr

content increases the austenitic dendrite fraction and reduces the fraction of $M_7C_3$ in the interdendritic regions.

ThermoCalc calculation in the given chemical composition may predict the phase evolution during solidification and cooling under equilibrium conditions. The solidified microstructure of the alloy is governed by the contents of C and Cr or their carbon equivalent even in the specified range. The prediction of the microstructure by ThermoCalc would be very useful in finding the microstructural evolution during and following heat treatment. Because the (dendritically) solidified austenite is supersaturated with C and Cr, the austenite would be in a metastable state during cooling below the solidus. The release of C and Cr from the supersaturated austenite and the precipitation of carbide could occur like the graphitization in gray iron [1].

In particular, the dendritically solidified austenite has a slightly different chemical composition from that of the eutectically solidified one. Thus, the fraction of dendritically solidified austenite in high Cr white iron also has a strong effect on its microstructural evolution and mechanical properties. The relationship between microstructural evolution during solidification or heat treatment and mechanical properties of high Cr white iron may be related to a carbon equivalent, however this is hard to find in previous studies.

In the present study, phase prediction through ThermoCalc calculation with various carbon equivalents, the effect of chemical composition on the solidified microstructure, the microstructural evolution during heat treatment, and the mechanical properties in the hypoeutectic high Cr white iron have been studied.

## 2. Experimental Procedure

### 2.1. Specimen Preparation

To develop the microstructure with various fractions of dendritically solidified austenite, the chemical compositions of the specimens were designed by variation of carbon equivalent (Ceq) from 3.3 to 4.3 whose compositions were in the range of 2.1–2.9 wt.% C, and 24.0–27.0 wt.% Cr. The composition range was selected to minimize precipitation of primary Cr carbide from the melt ahead of eutectic solidification. The chemical compositions of the specimens are given in Table 1. Phase prediction has been carried out with the chemical composition in Table 1 through commercial software ThermoCalc 2019b, on the basis of database DB TCFE9 for Steels/Fe-Alloys (V.9.1).

The specimens were prepared by induction melting of the master alloy followed by secondary melting and then casting to a Y-block mold and an investment casting mold, respectively.

The cast specimens were subjected to conventional heat treatment and modified heat treatment. The conventional heat treatment condition is composed of homogenization at 1065 °C for 4 h followed by air cooling and tempering at 500 °C for 4 h or 250 °C for 4 h.

**Table 1.** Chemical composition of the experimental high Cr white iron alloys (wt.%).

| Designation | Composition | | | | | |
|---|---|---|---|---|---|---|
| | C | Si | Mn | Cr | Ni | Mo |
| 2124 | 2.12 | 0.66 | 0.65 | 24.05 | 0.85 | 0.86 |
| 2127 | 2.13 | 0.71 | 0.67 | 27.00 | 0.89 | 0.86 |
| 2427 | 2.43 | 0.68 | 0.70 | 27.06 | 0.86 | 0.85 |
| 2827 | 2.78 | 0.70 | 0.70 | 27.34 | 0.87 | 0.85 |
| 2927 | 2.95 | 0.67 | 0.71 | 26.94 | 0.92 | 0.82 |

### 2.2. Microstructural Observation and Phase Identification

Specimens for optical microscopy (OM: ECLIPSE MA200, Nikon Co., Tokyo, Japan) and scanning electron microscopy (SEM: JSM–IT500LV, JEOL Ltd., Tokyo, Japan) were prepared metallurgically and etched by swabbing with Vilella's reagent consisting of

45 mL glycerol, 15 mL nitric acid (1.40), and 30 mL hydrochloric acid (1.19). Volume fraction of the primarily solidified dendrites in the as-cast specimens was measured during optical microscopy by Image Analyzer (Image&Microscope Technology/iSolution DT). The volume fraction measurement was carried out at the magnification of 50 with 10 views.

Specimens for transmission electron microscopy (TEM: JEM-2100F, JEOL Ltd., Tokyo, Japan) were prepared by mechanical polishing down to 50 micrometer thickness followed by twin jet polishing (Struers/TenuPol-5). The solution for twin jet thinning was 10% perchloric acid in methanol. The existing phases in the heat-treated specimens were identified by TEM selected area diffraction pattern (SADP), and energy dispersive X-ray spectrometer (EDS: X-Max$^N$, Oxford Instrument plc, Abingdon, UK).

### 2.3. Mechanical Properties

Wear tests of the specimens were conducted by a pin-type wear tester (UMT-TriboLab, Bruker, Billerica, MA, USA) with the dimension of 20 mm × 20 mm × 4 mm. The specimens were tested based on the basic conditions of ASTM G133 using linearly reciprocating ball-on-flat sliding wear geometry with a 6 mm zirconia ceramic ball. The testing force was 25 N and 30 N and the volume loss, damaged width, and damaged mean depth were measured at the traveling distance of 100 m, 500 m and 1000 m [3].

Rockwell Hardness values (WOLPERT/D-6700) for the specimens were measured in as-cast, conventionally heat-treated, and modified heat-treated conditions.

### 2.4. Directional Solidification and Quenching (DSQ)

To understand solidification behavior of the alloys, directional solidification and quenching (DSQ) experiments were conducted with a 5 mm diameter rod in an alumina tube at the solidification rate of 50 μm/s with a thermal gradient of 100 °C/cm at the solid/liquid interface. To preserve the liquid/solid interface, the alloy embedded alumina tube was suddenly dropped into the water bath located beneath the furnace during solidification. The quenched specimens were prepared along the solidification direction metallurgically and etched by swabbing with Vilella's reagent for metallography.

## 3. Results and Discussions

### 3.1. Prediction of the Microstructure

The excellent wear resistance of high Cr white iron is attributed to the existence of eutectic Cr carbides. The Cr carbides are hard, and most of the Cr carbides are $M_7C_3$ type eutectic carbides.

Pearce and Ohide found that with increasing Cr content the eutectic carbide changes from $M_3C$ to $M_7C_3$ [4,5]. Thus, the eutectic carbide in the present study of high Cr white iron is $M_7C_3$. However, the primary Cr carbides which precipitate from the melt ahead of eutectic solidification are deleterious to impact toughness and should be avoided in castings subjected to impact in service [2]. The composition range in Table 1 was selected to minimize precipitation of primary Cr carbide from the melt ahead of eutectic solidification [2]. Usually, high Cr white iron is produced as hypoeutectic compositions. As mentioned above, the solidified microstructure of the high white iron is governed by C and Cr contents, which means that it is also related to carbon equivalent (Ceq). Generally, eutectic composition of gray iron is 4.3 Ceq. For simple consideration, the 2927 alloy in Table 1 which has a fully eutectic microstructure, was counted as 4.3 Ceq. The Ceq of the chemical composition given in Table 1 was calculated with some modification. The Ceq value of each alloy is given in Table 2. The alloy with low Ceq value has high fraction of the primarily solidified dendrites similar to those in gray iron. The equilibrium phase fraction of each phase in the as-cast condition is also predicted through ThermoCalc calculation as shown in Figure 1 and Table 2.

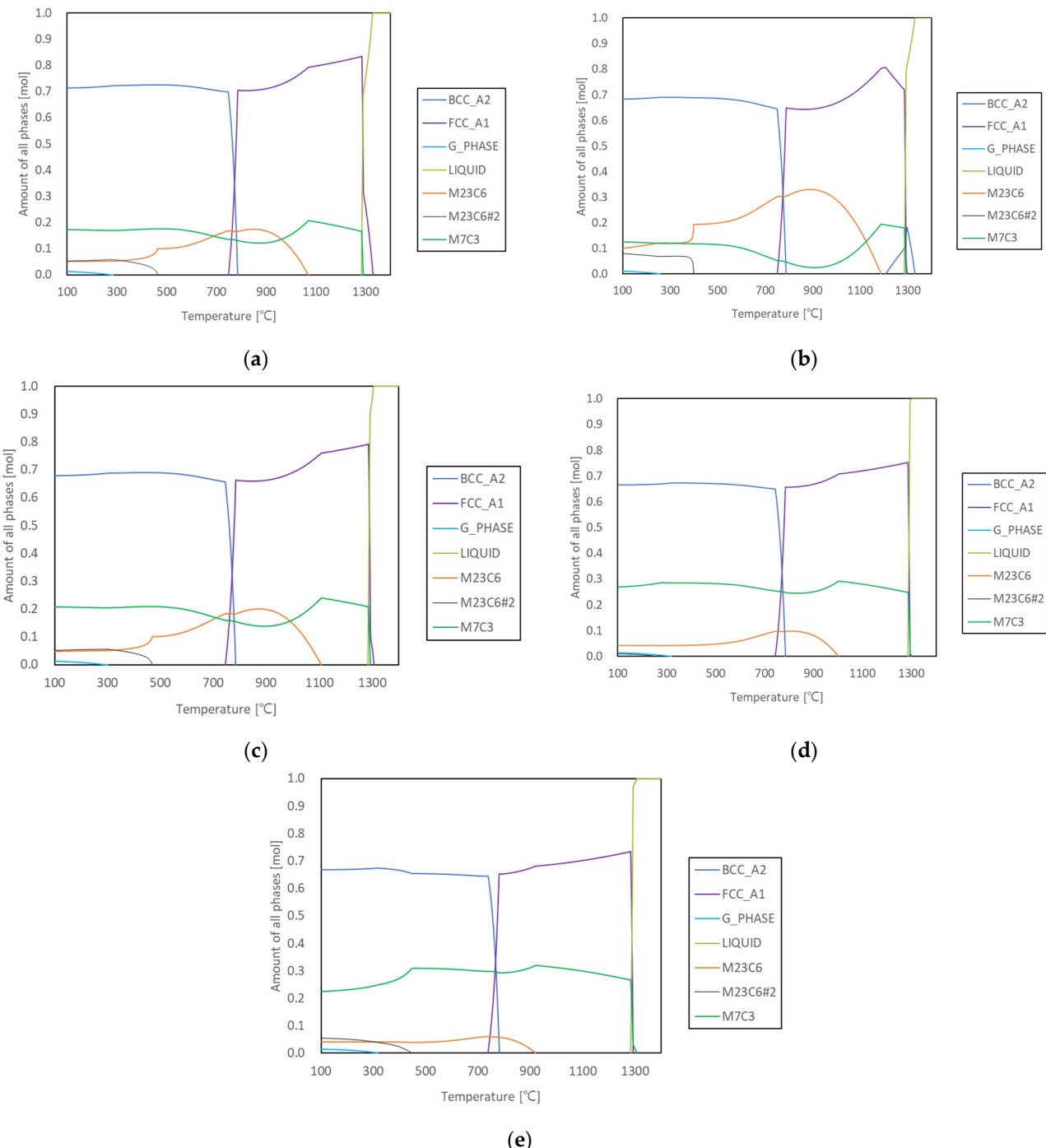

**Figure 1.** ThermoCalc calculation of the alloys. (**a**) 2124, (**b**) 2127, (**c**) 2427, (**d**) 2827, (**e**) 2927.

Solidification begins with austenite or delta ferrite (which will transform to austenite during subsequent cooling) in the relatively low Ceq alloys (2124, 2127 and 2427) at 1320 °C and 1300 °C, while addition of small fractions of $M_7C_3$, 0.1% in 2827 and 1.6% in 2927, led to solidification at 1300 °C ahead of eutectic reaction. Eutectic reaction begins at and around 1290 °C in all alloys. The alloy with fully eutectic structure 2927 shows the phase fraction of austenite to $M_7C_3$ is 2.76:1. From this phase fraction ratio, it is possible to calculate the primarily solidified austenite dendrites fraction (FPD) by subtraction of the fraction of $M_7C_3$ ($F_{M_7C_3}$) from the total austenite fraction (FA) at the final freezing temperature as following:

$$FPD = FA - 2.76 \times F_{M_7C_3} \tag{1}$$

where, the fraction of each phase means that at the final freezing temperature.

For example; the FPD of 2124 alloy would be $83.4 - 16.6 \times 2.76 = 37.6$ mol.%. The calculated fraction of the FPD for the alloys is displayed in Table 2. The remaining liquid fraction at the eutectic temperature is expected to contribute to the formation of a eutectic structure of austenite and $M_7C_3$ with the ratio of 2.76:1 at the final freezing temperature. This was confirmed with the microstructure of the fully eutectic alloy, 2927. However, 2127 alloy whose solidification begins with delta ferrite did not have such a relationship as in Equation (1). It is expected to originate from the different composition of delta ferrite from the austenite. The fraction of delta ferrite was impossible to measure by metallography and DSQ due to its early transformation to austenite (finally martensite or pearlite).

**Table 2.** ThermoCalc prediction of solidification behavior of the alloys (mol.%).

| Alloy | 2124 (Ceq:3.3) | 2127 (Ceq:3.5) | 2427 (Ceq:3.8) | 2827 (Ceq:4.1) | 2927 (Ceq:4.3) |
|---|---|---|---|---|---|
| Solidification leading phase and temperature (°C) | $\gamma$(8.8%) 1320 | $\delta^{*,1}$ (5.9%) 1320 | $\gamma$(5.4%) 1300 | $M_7C_3$ (0.1%) 1300 | $M_7C_3$ (1.6%) 1300 |
| Phase fraction ($\gamma$:L:$M_7C_3$) and eutectic reaction beginning temperature (°C) | 32.1:67.6:0.3 at 1290 | 37.6*,2:57.7:4.7 at 1290 | 39.1:52.5:8.4 at 1290 | 38.5:48.4:13.1 at 1290 | 26.7:62.0:11.3 at 1290 |
| Phase fraction ($\gamma$:$M_7C_3$) at final freezing temperature (°C) | 83.4:16.6 at 1284 | 81.9*,3:18.1 at 1285 | 79.1:20.9 at 1285 | 75.2:24.8 at 1285 | 73.4:26.6 at 1284 |
| Primarily solidified dendrite fraction | 37.6 | 31.9 | 21.4 | 6.8 | 0 |

*,1 Maximum delta ferrite fraction at 1296 °C was 18.2% then transformed gradually to austenite. *,2 This value includes 12.2% delta ferrite which will transform to austenite during subsequent cooling. *,3 This value includes 9.9% delta ferrite at the final freezing temperature.

### 3.2. As-Cast Microstructure

Figure 2 shows the as-cast microstructure of the alloys. As predicted in ThermoCalc calculation, the FPD increases with decreasing Ceq. The lowest Ceq alloy 2124 has a high fraction of dendrite, and little dendrite is found in the highest Ceq alloy 2927. The measured dendrite fractions of the alloys are listed in Table 3 in comparison with those of the ThermoCalc calculation.

**Table 3.** Measured dendrite fraction of the alloys.

| Alloy | 2124 (Ceq:3.3) | 2127 (Ceq:3.5) | 2427 (Ceq:3.8) | 2827 (Ceq:4.1) | 2927 (Ceq:4.3) |
|---|---|---|---|---|---|
| Measured dendrite fraction (vol.%) | 48.4 | 42.2 | 32.4 | 9.3 | 0 |
| ThermoCalc prediction (mol.%) | 37.6 | 31.9*,1 | 21.4 | 6.8 | 0 |

*,1 This value was derived by the calculation of Equation (1); where, the total austenite fraction at eutectic temperature was counted as the sum of delta ferrite, dendritic austenite, and austenite in eutectic.

As discussed above, the measured dendrite fractions have similar propensities to the calculated results that show dendrite fraction decreases with increasing Ceq, with the exception of the 2127 alloy, despite the unit of values being different from each other: mol. % and vol.%. The dendrite fractions between the measured values and the calculated ones have some difference because of different units. Because the calculated values are based on the equilibrium solidification and mol. fraction, the values might be different from those of the measured values whose solidification occurred under non-equilibrium conditions and vol.%. Though the values are hard to compare directly, they have similar propensities with the variations of Ceq.

The microstructure of the high Cr white iron is mainly composed of $M_7C_3$ carbide and austenite regardless of Ceq value. The eutectic composition 2927 alloy has fully eutectic structure of $M_7C_3$ and austenite (that transformed to martensite during post casting cooling) as shown in Figure 2e. In the low Ceq alloys, 2124, 2127 and 2427 have microstructures consisting of eutectic structures ($M_7C_3$ and austenite) with primarily solidified dendrites of austenite. In the 2124, 2127 and 2427 alloys the eutectic structure formed in the interdendritic region as displayed in Figure 2a–c. Incidentally, a small

fraction of dendrite is enveloped by the eutectic phase ($M_7C_3$ + austenite) in the 2827 alloy where eutectic structure was the leading phase in solidification as mentioned in Table 2 and shown in Figure 2d.

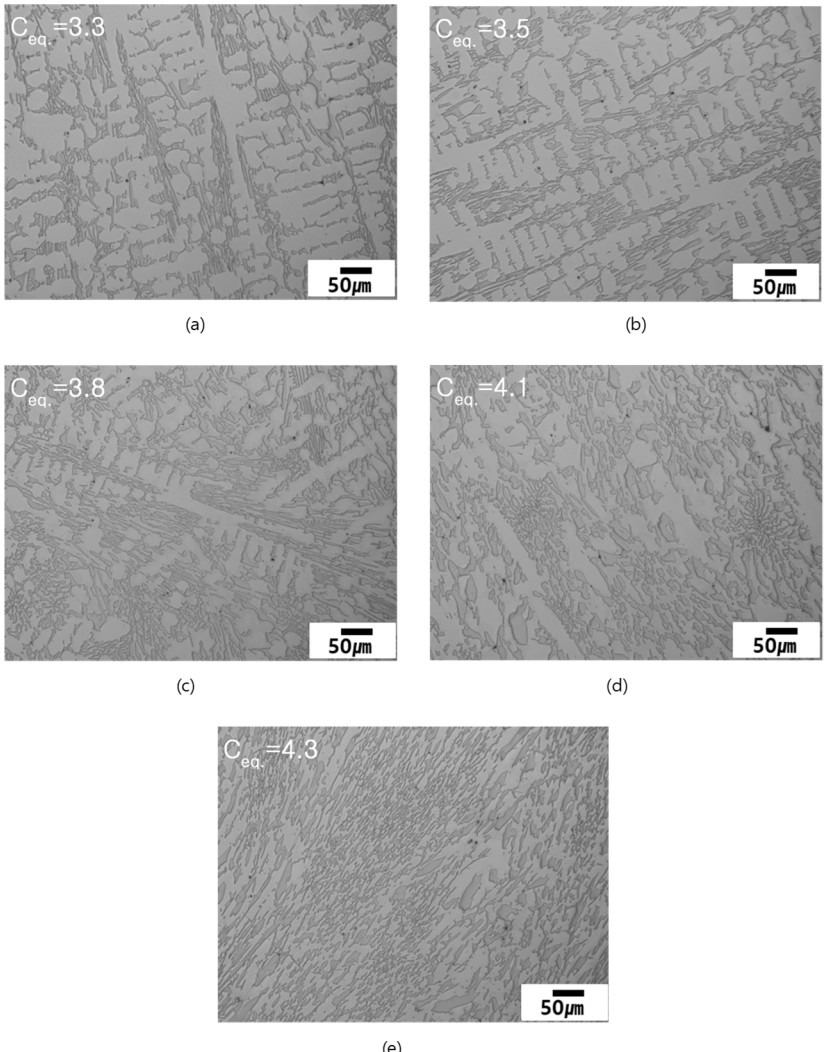

**Figure 2.** Optical micrograph of the alloys showing dendrite fraction decreases with increasing Ceq. (**a**) 2124, (**b**) 2127, (**c**) 2427, (**d**) 2827, (**e**) 2927.

### 3.3. Solidification Behavior of the Alloys

Solidification behavior of the alloys was studied by DSQ experiments. In the actual casting the solidification does not occur at a simple plane solid/liquid interface due to constitutional supercooling of the alloy. Generally, in dendritic solidification the preserved solid/liquid interface by DSQ has a certain range of solidification which is known as the mushy zone. It has been confirmed for each alloy that dendritic solidification occurred, as shown in Figure 3 which shows the solid/liquid interface of each alloy. The fully solidified region shows that dendrites have grown along the solidification direction for the alloys 2124, 2127, and 2427. This means that freezing of dendrites from the liquid was the leading phase of the solidification. Incidentally, the 2827 and 2927 alloys do not have any dendrite in the fully solidified region at the bottom of the mushy zone.

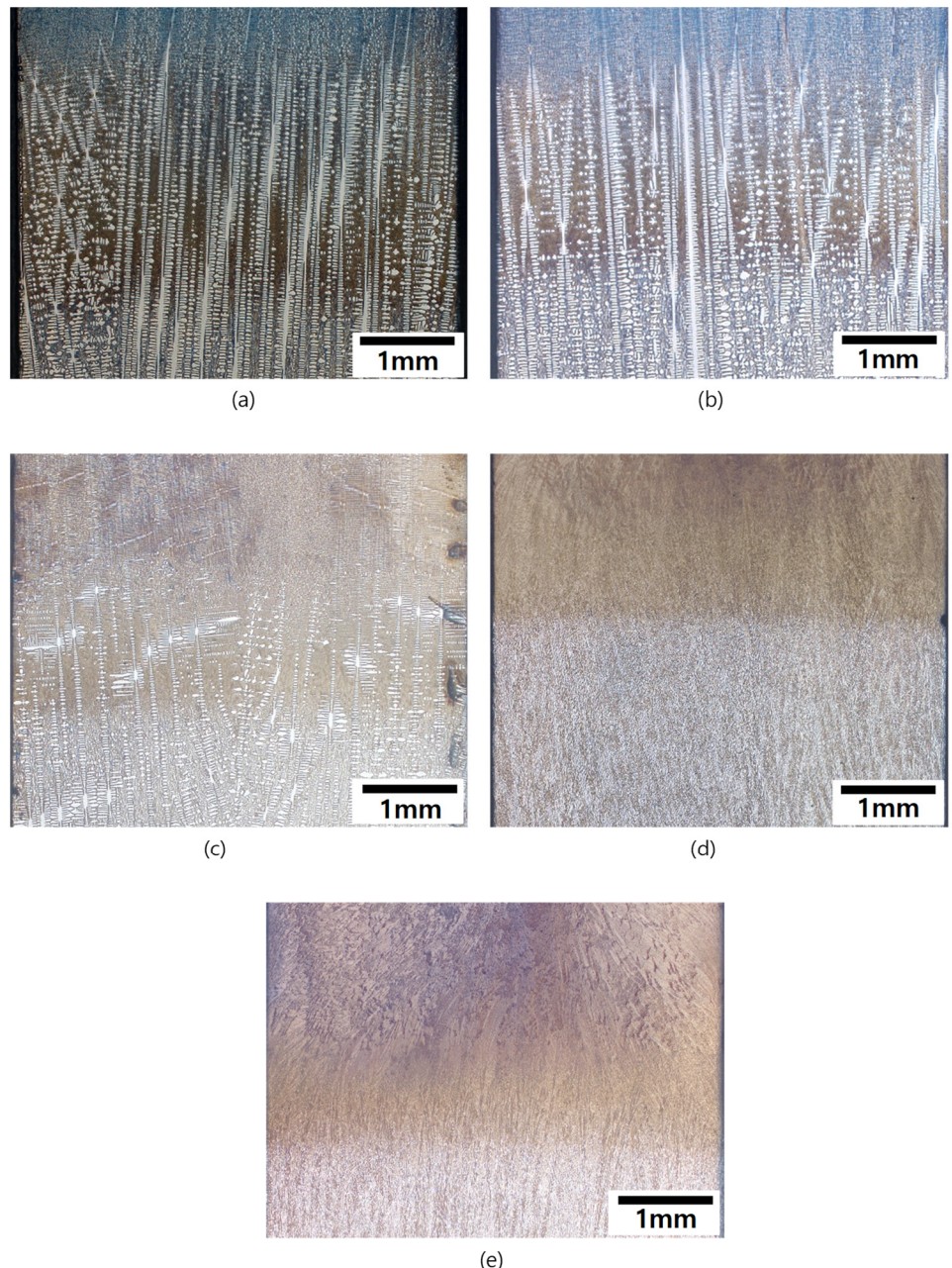

**Figure 3.** Optical micrographs along the solidified direction showing the preserved solid/liquid interface by DSQ. (**a**) 2124, (**b**) 2127, (**c**) 2427, (**d**) 2827, (**e**) 2927.

The micrographs perpendicular to the solidification direction at the fully solidified area (at the bottom of the mushy zone) are shown in Figure 4 showing dendritic solidification in low Ceq alloys, but its absence in the 2827 and 2927 alloys. It is natural that the eutectic alloy 2927 is dendrite free, however, the 2827 alloy whose calculated dendrite fraction is 6.8, does not have any dendrite as shown Figure 4. This is expected to be caused by the steady-state solidification characteristics during directional solidification. At the transient stage of the solidification the solute concentration begins to increase, then the solute concentration becomes constant in the steady state, and finally the solute elements become rich at the final stage of directional solidification. Thus, in the eutectic or near eutectic alloy it is impossible for the primary dendrite to appear during directional solidification as displayed in Figures 3d,e and 4d,e [6].

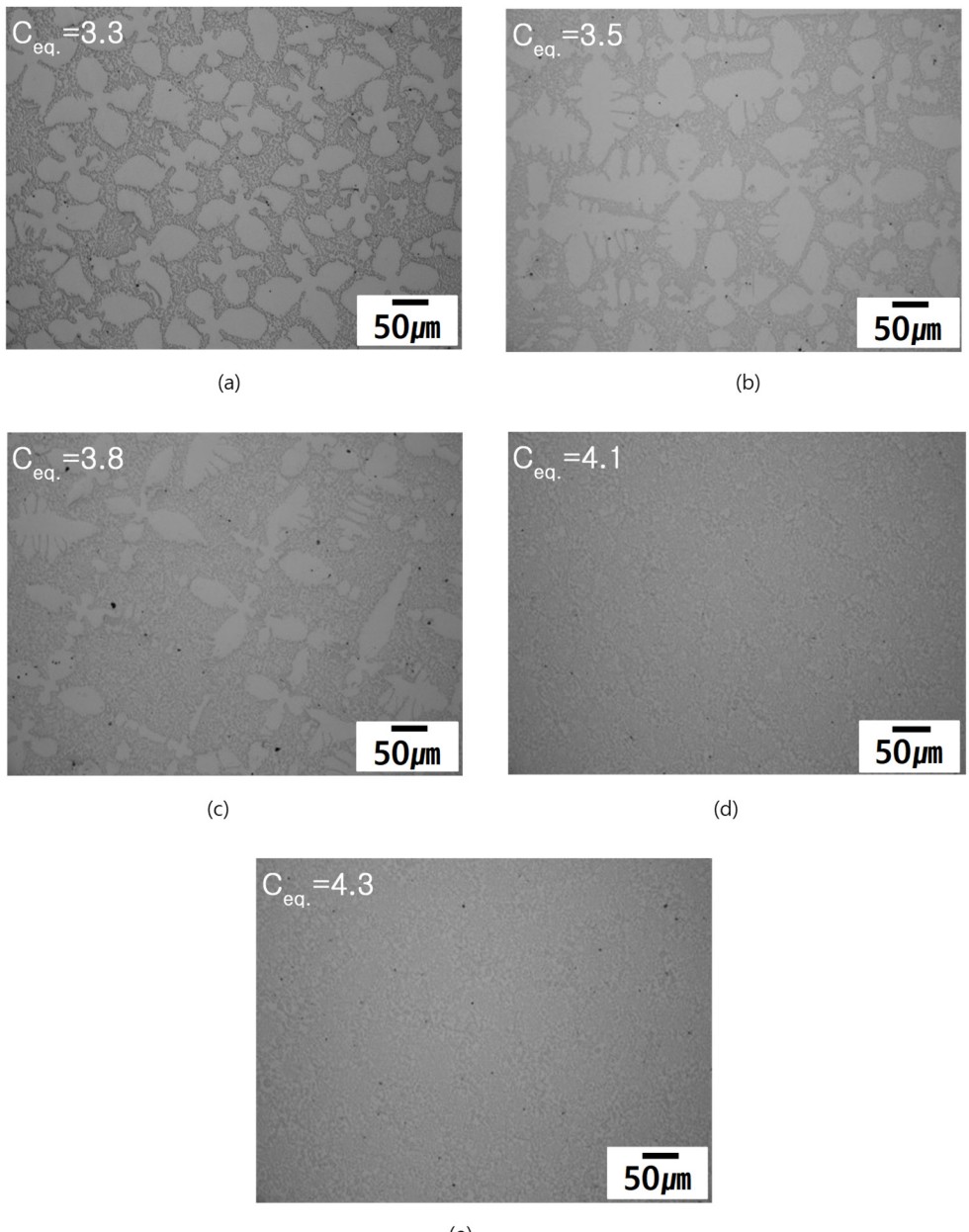

**Figure 4.** Cross sectional optical micrographs at the bottom of the mushy zone where full solidification occurred during DSQ: (**a**) 2124, (**b**) 2127, (**c**) 2427, (**d**) 2827, (**e**) 2927.

### 3.4. Heat Treated Microstructure

#### 3.4.1. Microstructural Evolution during Heat Treatment

The specimens were subjected to various heat treatments as following:

- Destabilizing and low-temperature aging 1065 °C for 4 h AC (Air cooling to room temperature) + 500 °C for 4 h (below: DES + Age) 1065 °C for 4 h WQ (water quench) + 500 °C for 4 h (below: DES + WQ + Age);
- Destabilizing and low-temperature tempering 1065 °C for 4 h AC + 250 °C for 4 h (below: DES + Temper);
- Modified destabilizing and low-temperature tempering 1065 °C for 1 h AC + 250 °C for 1 h (below: MDES + Temper).

Conventional heat treatment of the high Cr cast iron involves heating and holding at soaking temperature (usually 950–1065 °C), followed by air cooling and tempering as with DES + Temper. The holding or soaking duration is referred to as 'destabilization'

since it allows the solute elements such as C and Cr to come out from the supersaturated austenitic matrix to form secondary carbides. The $M_{23}C_6$ carbide precipitation from the supersaturated austenite during destabilization is expressed as following in the previous report [7]:

$$\gamma \ \rightarrow \ \gamma^* + M_{23}C_6 \tag{2}$$

where, $\gamma^*$ is austenite with lower alloy content than that of the original matrix $\gamma$ [8].

Figure 5 shows the microstructure of the destabilized and low-temperature aged (DES + Age) alloys. The micrographs of the DES + Age treated specimens have similar dendrite feature to those of the as-cast, but dendritic boundaries were not distinct compared with those of the as-cast ones in Figure 2. This might be caused by the destabilization at high temperature which accelerates diffusion of the alloying elements and carbide precipitation. Compared with those of the as-cast, the dendrites are filled with many particles in the DES + Age treated specimens.

SEM micrographs of the as-cast and the DES + Age treated specimens are displayed in Figures 6 and 7 respectively. Little or no particles exist within the dendrites in the as-cast as displayed in Figure 8, whereas precipitates are abundant in the DES + Age treated condition. As predicted through ThermoCalc calculation, the particles are expected to be secondary $M_{23}C_6$ carbide. TEM micrographs of the representative 2427 alloy reveal that the particles are $M_{23}C_6$ carbide as shown in Figure 9.

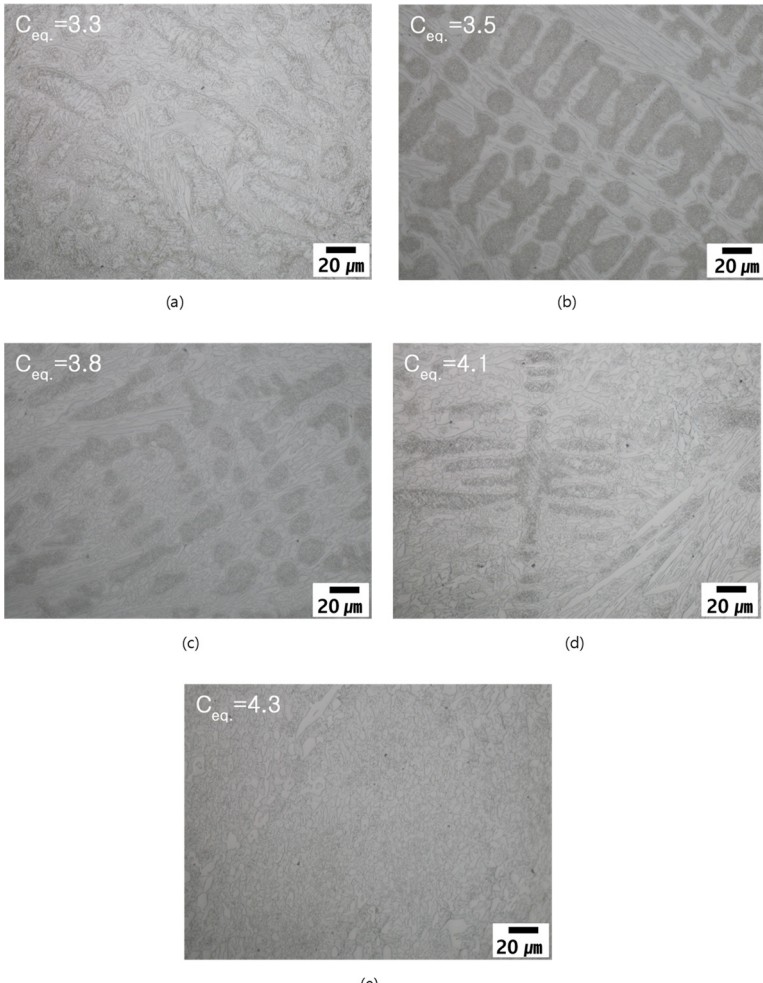

**Figure 5.** Optical micrographs of the heat treated (DES + Age) alloys showing dendrite fraction decreases with increasing Ceq; (**a**) 2124, (**b**) 2127, (**c**) 2427, (**d**) 2827, (**e**) 2927.

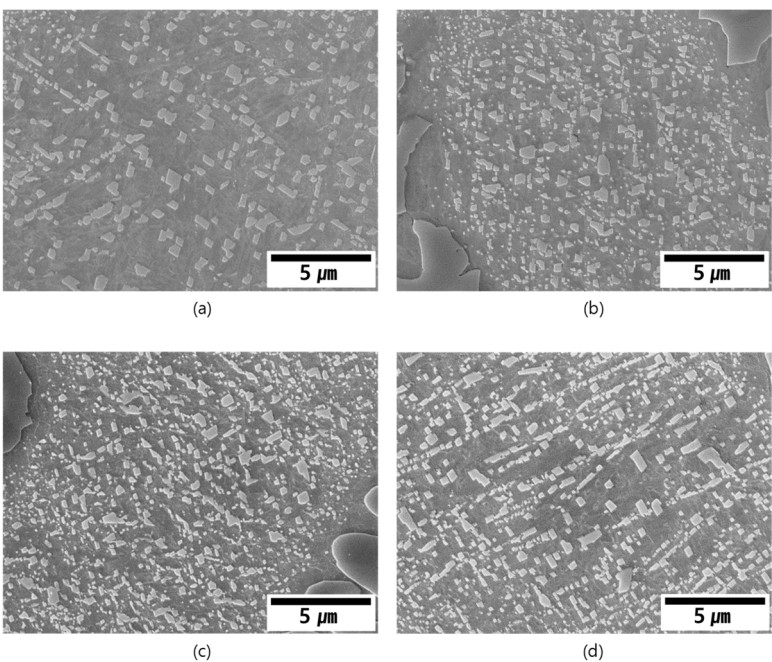

**Figure 6.** SEM micrographs in the dendrite area of the DES + Age treated alloys; (**a**) 2124, (**b**) 2127, (**c**) 2427, (**d**) 2827; where 2927 alloy was excluded due to absence of dendrite.

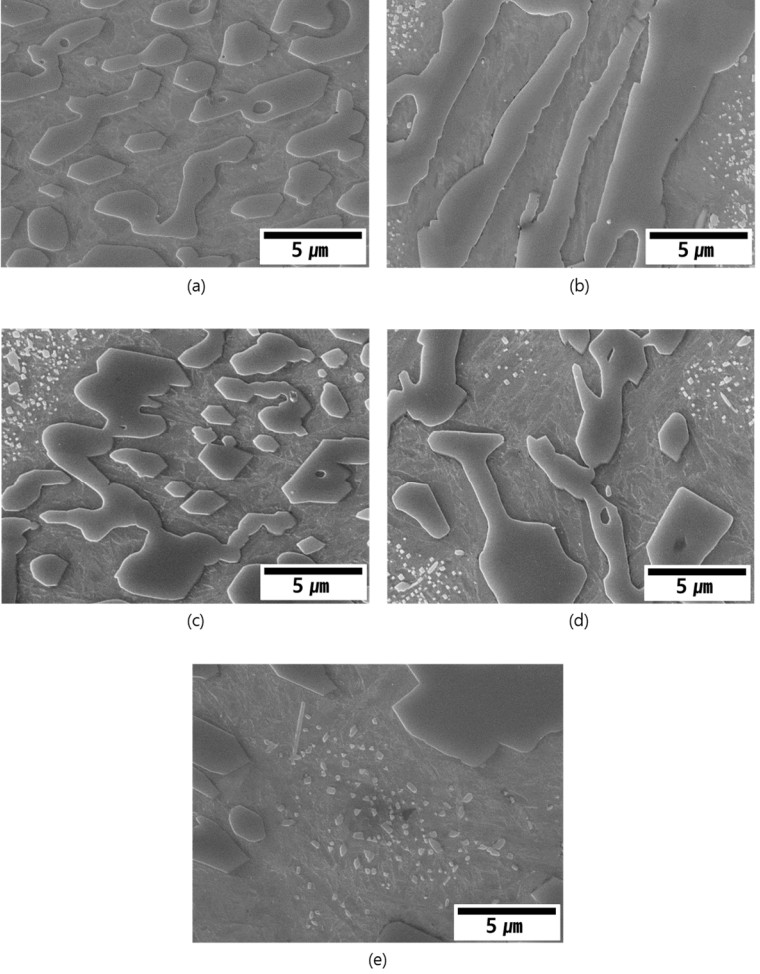

**Figure 7.** SEM micrographs of interdendritic eutectic region in the DES + Age treated alloys; (**a**) 2124, (**b**) 2127, (**c**) 2427, (**d**) 2827, (**e**) 2927.

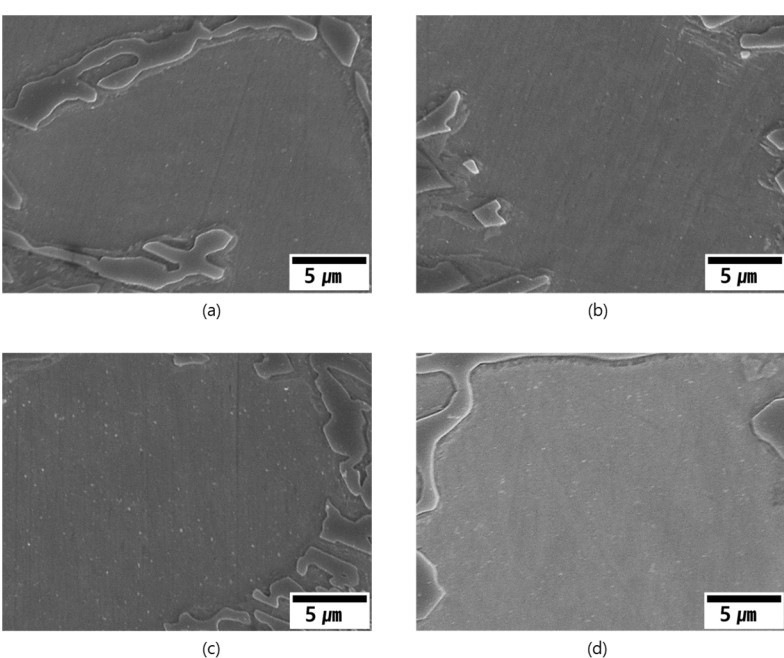

**Figure 8.** SEM micrographs in the as-cast alloys showing little or no precipitation within dendrite; (**a**) 2124, (**b**) 2127, (**c**) 2427, (**d**) 2827.

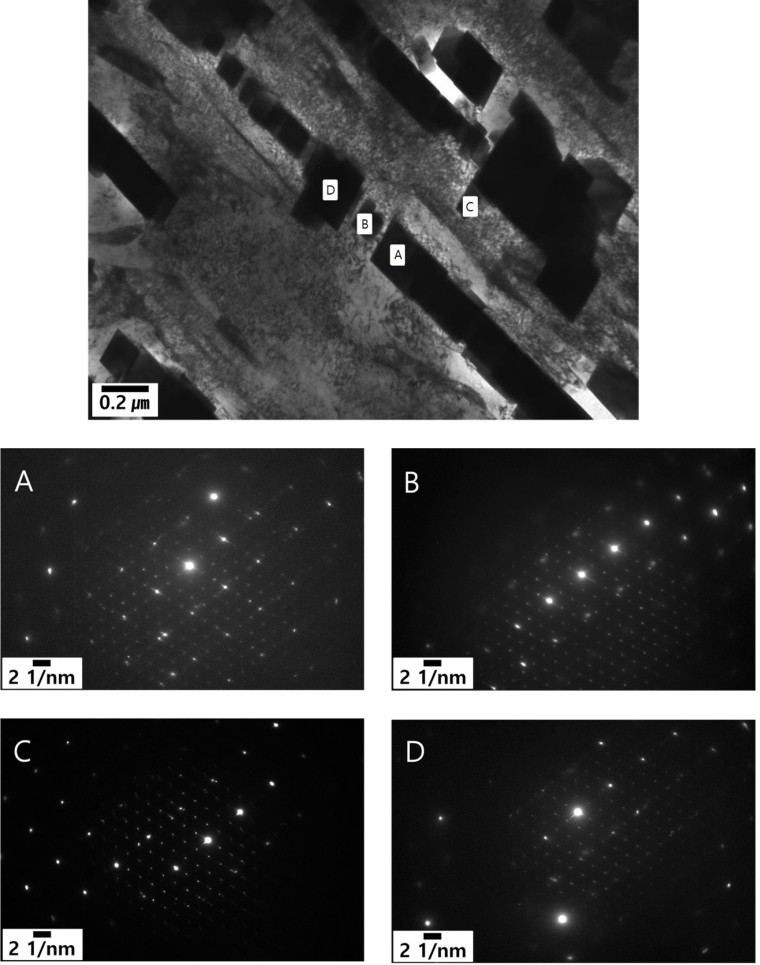

**Figure 9.** TEM micrograph in the dendritic region of the DES + Temper treated 2427 alloy showing $M_{23}C_6$ SADP patterns at each point(**A–D**).

Though precipitation of the particles during DES + Age treatment within the dendrites was abundant, the interdendritic eutectic region did not have such an amount of particles as shown in the SEM micrographs of Figure 7. Excepting the 2927 alloy (eutectic alloy), precipitation of the particles scarcely occurred within the austenite in the eutectic alloys. Many studies also report that the secondary carbides do not nucleate and grow on the eutectic carbides but form preferentially within dendritic matrix [9]. They offered two explanations for this phenomenon [9]:

1. Formation of $M_{23}C_6$ as a shell of $M_7C_3$ in the eutectic structure is a peritectoid type reaction between $M_7C_3$ and austenite matrix.
2. $M_7C_3$–$M_{23}C_6$ transition and precipitation of $M_{23}C_6$ from the adjacent austenite matrix.

This was similar to the present study for the hypoeutectic alloys as displayed in Figures 6 and 7. By comparison with Figures 6 and 7, the precipitation is strongly related to the existence of the primarily solidified dendrites in the hypoeutectic alloys as reported earlier [9]. This means that the primarily solidified dendrites have high potential to form precipitates due to supersaturation of solute elements such as Cr and C. However, in the 2927 alloy, which has a fully eutectic structure in the absence of the primarily solidified dendrite, the solid-state precipitation of $M_{23}C_6$ carbides occurred slightly in the austenite (which transformed to martensite during subsequent cooling) and solidified during the eutectic reaction as displayed in Figure 7e. A little precipitation occurred in the austenite in eutectic alloys which have a relatively wide gap among the $M_7C_3$ particles.

### 3.4.2. Effect of Heat Treatment on $M_{23}C_6$ Precipitation

As mentioned above, the alloys were subjected to various heat treatments.

1. Effect of low temperature aging (DES + Age) and tempering (DES + Temper).

Conventional heat treatment of the high Cr cast iron is known as destabilizing, and undertaken at 1065 °C for 4 h followed by air cooling (AC) and tempering at 250 °C for 4 h (DES + Temper) [2]. The heat-treated microstructure of the alloy is expected to be composed of martensite (originated from the primarily solidified and eutectically solidified austenite) matrix, $M_7C_3$, and $M_{23}C_6$. The matrix (austenite: transformed to martensite) and the $M_7C_3$ carbide originate from the solidification regardless of composition or dendrite volume fraction. However, $M_{23}C_6$ carbide is a solid-state precipitation product during heat treatment or cooling. In Figure 10, images of the DES + Temper treated alloys show $M_{23}C_6$ carbide particles precipitated in the dendrite area. The amount of precipitation is different from each alloy. $M_{23}C_6$ precipitation in 2127 and 2427 alloys (Figure 7b,c) occurred a bit more than that in 2124 and 2827 (Figure 7a,d) alloys as listed in Table 4 of ThermoCalc prediction.

**Table 4.** Precipitation behaviors of $M_7C_3$ and $M_{23}C_6$ by ThermoCalc calculation(mol.%).

| Alloy | $M_{23}C_6$ Precipitation Initiation | | $M_{23}C_6$ Peak Precipitation | | |
|---|---|---|---|---|---|
| | T (°C) | $M_7C_3$ Fraction | T (°C) | $M_{23}C_6$ Fraction | $M_7C_3$ Fraction |
| **2124** | 1060 | 19.8 | 850 | 17.3 | 12.2 |
| **2127** | 1180 | 18.8 | 890 | 33.0 | 2.5 |
| **2427** | 1100 | 23.4 | 870 | 20.1 | 14.0 |
| **2827** | 1000 | 29.1 | 810 | 9.8 | 24.6 |
| **2927** | 920 | 31.9 | 739 | 6.0 | 29.7 |

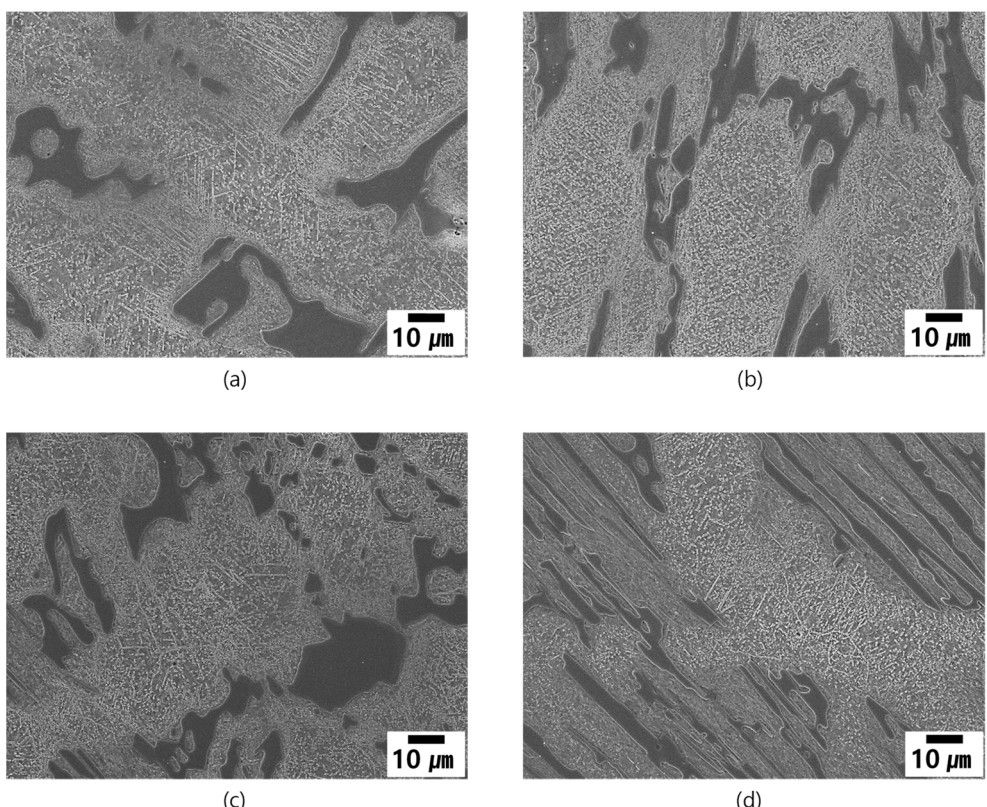

**Figure 10.** SEM micrographs of a dendrite region of the DES + Temper treated alloys; (**a**) 2124, (**b**) 2127, (**c**) 2427, (**d**) 2827; where 2927 alloys were excluded due to absence of dendrite.

$M_{23}C_6$ precipitation in the interdendritic region of Figure 7 displays clear differences among the specimens. $M_{23}C_6$ carbide particles precipitated scarcely in the interdendritic regions (or eutectic regions) of 2124, 2827, and 2927 alloys, and similar precipitation occurred in 2127 and 2427 alloys. The microstructural difference among the alloys with DES + Temper treatment can be understood by the careful analysis of ThermoCalc curves in Figure 1. $M_{23}C_6$ precipitation begins at 1060 °C in 2124, 1180 °C in 2127, 1100 °C in 2427, 1000 °C in 2827, and 920 °C in 2927. The destabilizing at 1065 °C for 4 h assisted precipitation of $M_{23}C_6$ carbide in the dendrites especially in 2127 and 2427, and also slightly assisted it in 2124 and 2827, whose precipitation temperature was a bit lower. This is expected to originate from microsegregation within the dendrite, which has a difference in solute concentrations from the dendrite core to the periphery of dendrite as reported in previous studies [9]. Microsegregation might lead to different destabilizing temperatures of $M_{23}C_6$ from the core to the periphery of the dendrite. Thus, some $M_{23}C_6$ carbide precipitation began at an even lower temperature than the calculated one. The precipitation was expected to be assisted by both the microsegregation and also heating to a destabilization temperature that passes through the temperature range of the maximum fraction temperature region in the ThermoCalc calculation of Figure 1.

In particular, the fraction of $M_{23}C_6$ was high in the 2127 alloy. This phenomenon might be caused by the peculiar solidification sequence as mentioned above, precipitation from the supersaturated solid, and reaction between $M_7C_3$ and matrix which occurred during heat treatment.

Micrographs of the DES + Age treated alloys are displayed in Figure 11. Comparison of micrographs between the DES + Age and the DES + temper for each alloy does not show much difference between Figures 10–12. This means that precipitation of $M_{23}C_6$ during destabilization is dependent upon neither tempering temperature nor aging temperature, but is governed by the destabilizing temperature. The heat treatments were conducted at the same destabilizing temperature. Thus, both heat treatments resulted in similar features of $M_{23}C_6$ precipitation for each alloy.

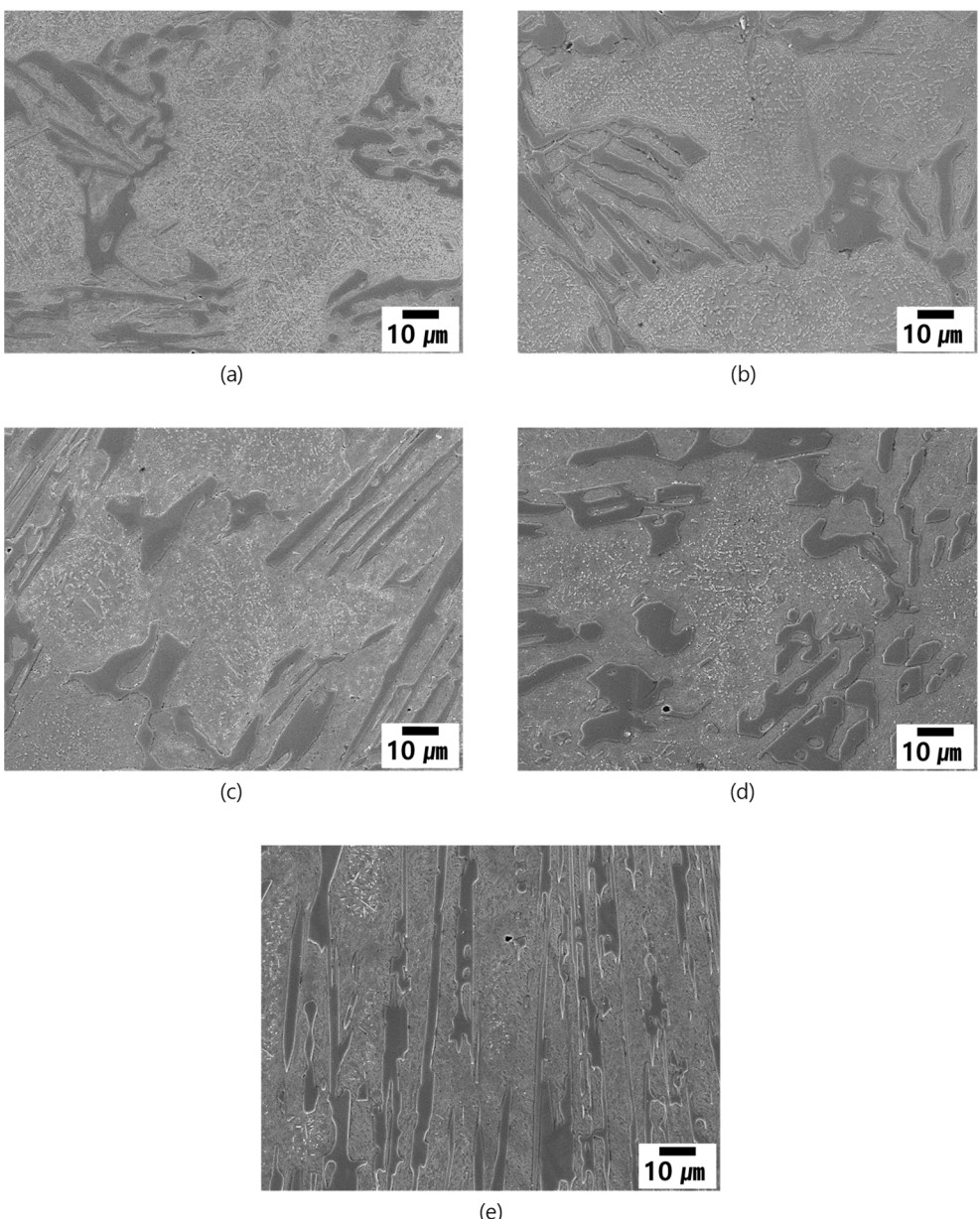

**Figure 11.** SEM micrographs of DES + Age treated (aged at 500 °C) alloys; (**a**) 2124, (**b**) 2127, (**c**) 2427, (**d**) 2827, (**e**) 2927.

2. Effect of cooling condition

Micrographs of DES + WQ + Age (1065 °C destabilizing followed by WQ and low-temperature aging) specimens have similar features as those of DES + Age and DES + Temper specimens as displayed in Figure 13. The heat treatments were conducted at the same destabilization condition, but the cooling conditions and tempering (or aging) conditions were different from each other. Even though those heat treatment conditions were different from each other, the microstructural features of each alloy in Figure 13 are similar to those of each alloy in Figures 10–12. This demonstrates that the $M_{23}C_6$ precipitation is governed by the destabilizing treatment regardless of cooling or tempering conditions.

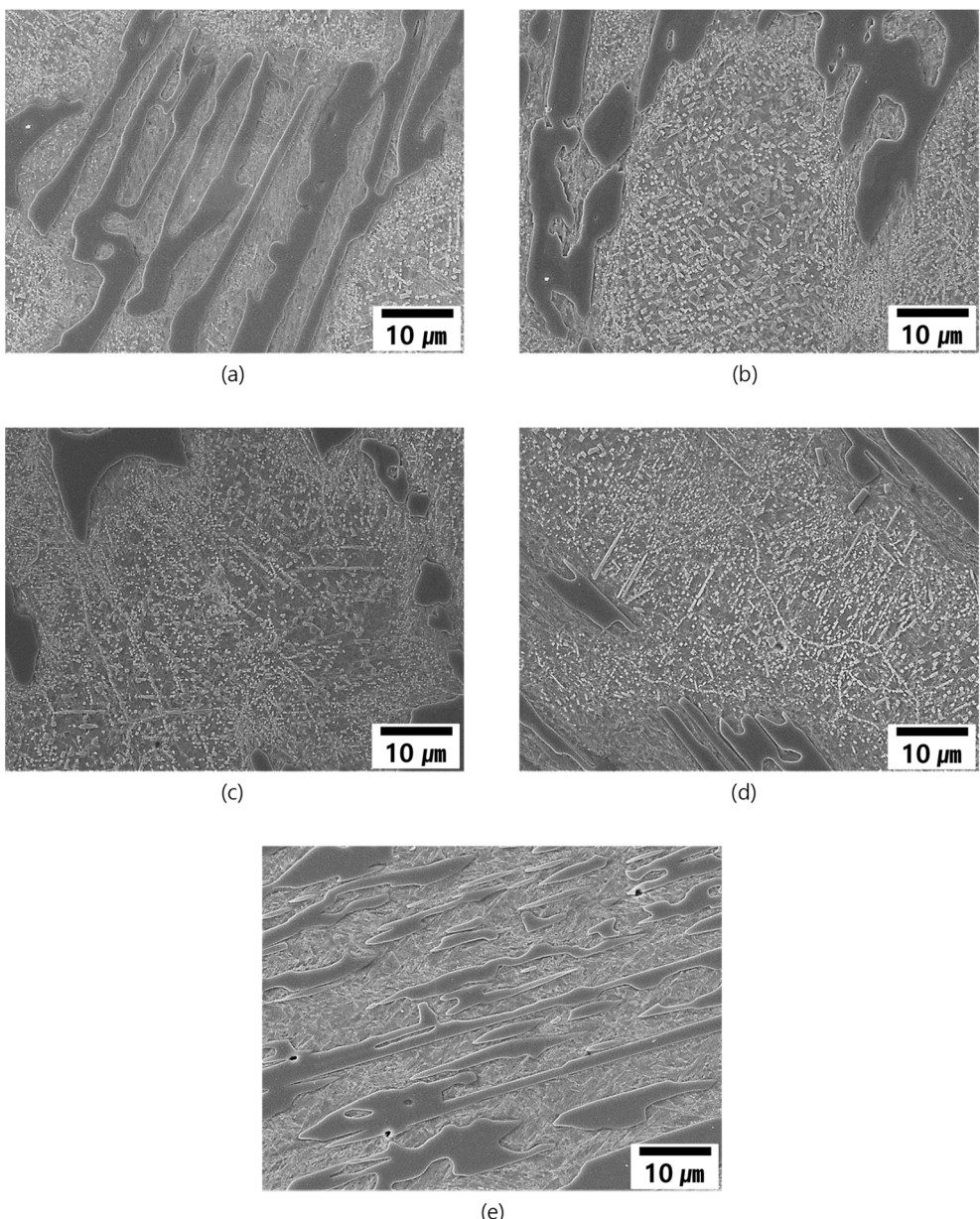

**Figure 12.** SEM micrographs of DES + Temper treated (conventionally heat treated) alloys showing difference of $M_{23}C_6$ precipitation in the interdendritic region depending on the alloys; (**a**) 2124, (**b**) 2127, (**c**) 2427, (**d**) 2827, (**e**) 2927.

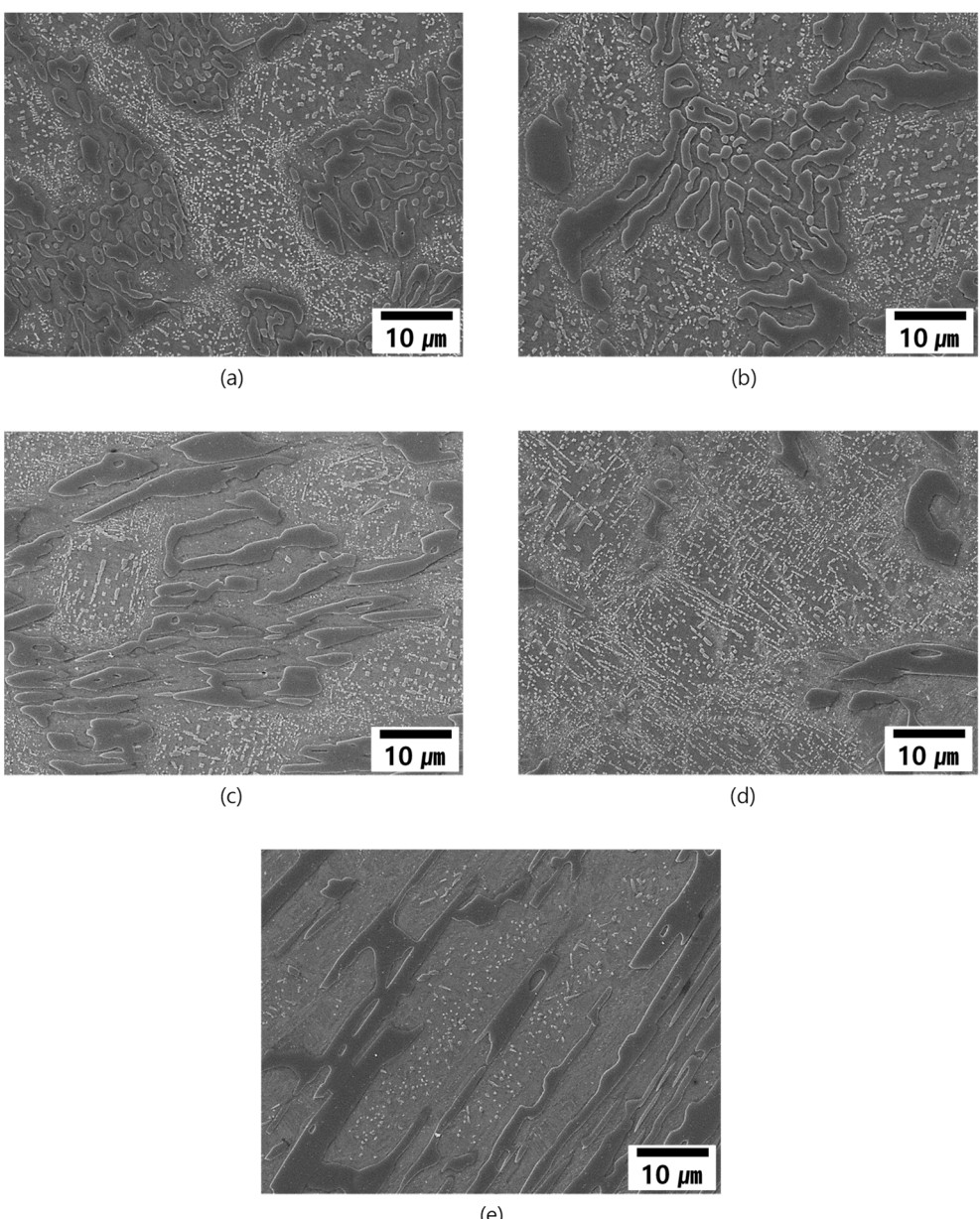

**Figure 13.** SEM micrographs of DES + WQ + Age treated alloys showing similarities to those in the DES + Age treated alloys; (**a**) 2124, (**b**) 2127, (**c**) 2427, (**d**) 2827, (**e**) 2927.

## 4. Mechanical Properties

### 4.1. Hardness

The hardness values of the alloys in the as-cast condition increased as the Ceq value was raised due to the high fraction of eutectic structure ($M_7C_3$ and austenite). Hardness increased with heat treatments in all alloys. The hardness value increment by heat treatment in the high dendrite fraction (low Ceq) alloys was more than that of the low dendrite fraction alloys (high Ceq or eutectic alloy) as shown in Table 5. As mentioned above, the hardness value of the high Cr white iron in the as-cast condition is closely related to the fraction of $M_7C_3$ carbide, thus the eutectic composition alloy, 2927, has the highest hardness among the alloys. Most of the alloys in the heat-treated conditions have similar hardness. Thus, the increment of hardness with heat treatment in the high dendrite fraction alloy, 2124, which had had the lowest hardness value in the as-cast condition, was greater than the other alloys. The increment in hardness with heat treatment may be caused by not only the transformation of austenite to martensite but also the precipitation of $M_{23}C_6$

carbide in dendritic austenite. In particular, the alloys with a high fraction of dendrite have high potentials of phase transformation and fine $M_{23}C_6$ precipitation. As appears in Figures 6, 10, 11 and 13, the alloys with high dendrite fractions have more $M_{23}C_6$ particles within the dendrites. The particles in the DES + WQ + Age specimen are relatively coarser than in other specimens. Thus, the hardness of the DES + WQ + Age specimen is a bit lower than other specimens. This is expected to be related to the nucleation and coarsening of the precipitates during thermal treatment.

**Table 5.** Effect of heat treatment on the hardness values of the alloys ($H_R$C).

| Alloy | 2124 | 2127 | 2427 | 2827 | 2927 |
|:---:|:---:|:---:|:---:|:---:|:---:|
| As-Cast | 48.0 | 46.8 | 48.0 | 50.3 | 49.8 |
| DES + Age | 60.0 | 59.3 | 59.7 | 59.3 | 60.5 |
| DES + WQ + Age | 52.3 | 57.7 | 57.7 | 58.3 | 61.3 |
| DES + Temper | 58.3 | 60.0 | 59.7 | 60.3 | 62.2 |
| MDES + Temper | 57.3 | 58.3 | 59.2 | 60.3 | 62.0 |

*4.2. Wear Resistance*

The volumetric wear loss during testing with a reciprocation ball on flat sliding for the DES + Temper alloys was closely related to the fraction of $M_7C_3$. The alloys with high fractions of $M_7C_3$ have better wear resistance than that of the alloys with low fractions of $M_7C_3$ as shown in Table 6. The volume loss decreases with increased fraction of $M_7C_3$ as listed in Table 4. Figure 14 shows that $M_7C_3$ prohibits matrix wear loss. The wear loss of the alloys with a high dendrite fraction showed more than the low dendrite fraction alloys, with deep water traces particularly visible in Figure 14a,b. The volumetric losses of the DES + Age (1065 °C 4 h + 500 °C 4 h aging) specimen were less than those of the MDES + Temper (1065 °C 1 h + 250 °C 1 h tempering) specimens. The hardness values of the DES + Age specimens are slightly lower than those of the DES + Temper specimens as shown in Table 5. Generally, wear resistance is closely related to the hardness value of the material. The good wear resistance of the DES + Age specimens is expected to originate from the higher fractions of $M_7C_3$ and $M_{23}C_6$ carbide precipitation.

**Table 6.** Reciprocation ball on flat sliding wear test (Volume loss, mm$^3$) for wear resistance under the test condition of load:25 N, frequency:5 Hz, distance: 1000 m, temperature: 24–25 °C, humidity: <60% RH.

| Alloy | 2124 | 2127 | 2427 | 2827 | 2927 |
|:---:|:---:|:---:|:---:|:---:|:---:|
| DES + Age | 0.352 | 0.367 | 0.338 | 0.306 | 0.297 |
| MDES + Temper | 0.239 | 0.281 | 0.201 | 0.271 | 0.256 |

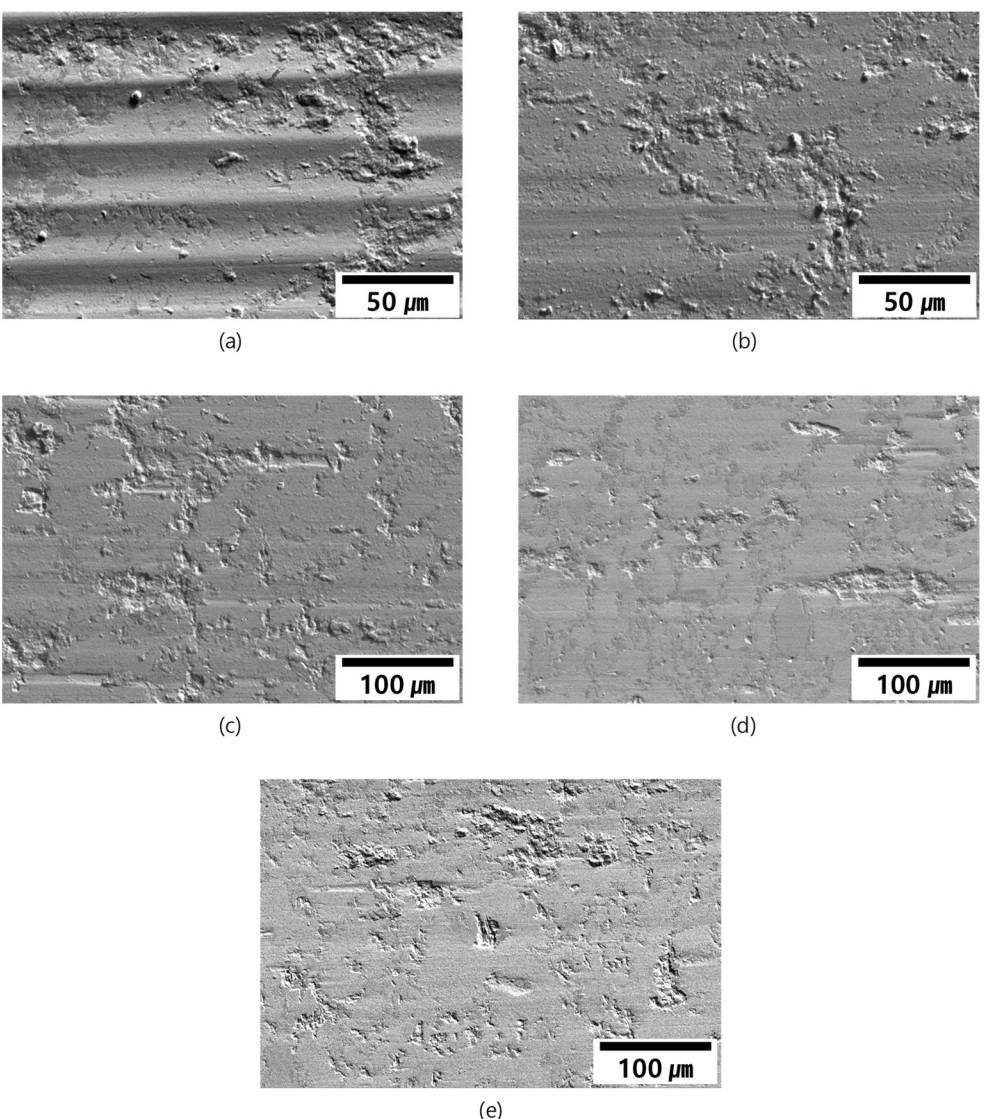

**Figure 14.** SEM micrographs of MDES + Temper treated alloys showing wear surface depending on the alloys; (**a**) 2124, (**b**) 2127, (**c**) 2427, (**d**) 2827, (**e**) 2927.

## 5. Conclusions

The effect of primary dendrite volume fraction through modification of chemical composition in high Cr white iron has been studied and reviewed. The summary is as follows:

- With increasing Ceq the fraction of the primarily solidified dendrites decreased. The measured and calculated fractions of the primarily solidified dendrites have similar propensity to Ceq;
- Eutectic reaction occurred with the ratio of $M_7C_3$:austenite = 1:2.76, which was predicted by ThermoCalc calculation;
- It was found that destabilization of austenite during conventional heat treatment releases the saturated solute elements C and Cr to form $M_{23}C_6$ in the primary (austenite) dendrite, however little or no $M_{23}C_6$ precipitation occurred within austenite in eutectic;
- The amount of $M_{23}C_6$ precipitation during destabilization is closely related to that of the primarily solidified dendrites which means that it is very sensitive to chemical composition;

- The mechanical properties such as hardness and wear resistance of the alloys are closely related to the fractions of $M_7C_3$ and $M_{23}C_6$.

**Author Contributions:** Conceptualization, C.-Y.J. and J.-H.L.; methodology, C.-Y.J. and J.-H.L.; software, B.-G.C.; validation, J.-S.O., Y.-G.S. and R.N.; formal analysis, J.-S.O., Y.-G.S. and C.B.; investigation, C.-Y.J.; resources, C.-Y.J. and J.-H.L.; data curation, J.-S.O., Y.-G.S. and C.B.; writing—original draft preparation, J.-S.O., Y.-G.S. and C.-Y.J.; writing—review and editing, J.-S.O., Y.-G.S., C.-Y.J. and J.-H.L.; visualization, J.-S.O.; supervision, C.-Y.J. and J.-H.L.; project administration, C.-Y.J. and J.-H.L.; funding acquisition, J.-H.L. All authors have read and agreed to the published version of the manuscript.

**Funding:** The APC was Funded by voucher of "Chang-Yong Jo".

**Institutional Review Board Statement:** Not applicable.

**Informed Consent Statement:** Not applicable.

**Data Availability Statement:** All the results are available on request to the corresponding author.

**Conflicts of Interest:** The authors declare no conflict of interest.

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
