# Peer review of "Effect of Dendrite Fraction on the M23C6 Precipitation Behavior and the Mechanical Properties of High Cr White Irons"

_metals, doi:10.3390/met11101576_

Round 1
Reviewer 1 Report
The manuscript describes a systematic study of as cast white iron as a function of composition and decomposition during heat treatment. This is an important topic, well done and well presented in general. There are, however, some shortcoming that should be overcoming before publication
Figure 1 is quire standard and can be removed
Line 145: ThermoCalc does not predict microstructures, only fractions. Also mention the data base used in the calculations.
Line 178: please comment how you determine volume fractions from 2D micrographs.
Line 232: I would prefer the expression “Austenite decomposition” instead of “destabilizing”.
Conclusion line 404 reformulate. What do you mean by “Eutectic reaction? How is eutectic solidification related to the fraction of M23C6?
General: replace “dendrite austenite” by “primary (austenitic) dendrite”, “eutectic austenite” by “austenite in eutectic”
Line 411: “it is very keen to chemical composition” à “it is very sensitive to chemical composition”
Line 413: In which way relate the mechanical properties to the composition? Please be more specific.
Author Response
Response to Reviewer 1 Comments
-Thank you for the kind and detailed comments!
Point 1: Figure 1 is quire standard and can be removed
Response 1: Deleted Figure 1. (p.6, line.148-149)
Point 2: Line 145: ThermoCalc does not predict microstructures, only fractions. Also mention the data base used in the calculations.
Response 2: Revised in the text: ‘The phase fraction of each phase in the as-cast condition---’ (p.7, line. 168-170)
And ‘ Data base is mentioned in the experimental procedure.’(p.3, line. 105-106)
Point 3: Line 178: please comment how you determine volume fractions from 2D micrographs.
Response 3: Mentioned in the experimental procedure (p.5, line. 122-123)
Point 4: Line 232: I would prefer the expression “Austenite decomposition” instead of “destabilizing”.
Response 4: Sorry that ‘destabilizing or destabilization in the alloy’ refers the decomposition of austenite and precipitation of M23C6, and it is generally accepted terminology.
Point 5: Conclusion line 404 reformulate. What do you mean by “Eutectic reaction? How is eutectic solidification related to the fraction of M23C6?
Response 5: Solidification of the hypoeutectic alloys begins with dendrite formation (austenite or delta ferrite), and followed by eutectic phase (austenite + M7C3). The eutectic reaction means formation of eutectic phase (austenite +M7C3) but does not related to M23C6 fraction in the alloys. Thus, the eutectic solidification (Austenite +M7C3) gives little effect on the fraction of M23C6. During post cooling M23C6 fraction gradually increase due to destabilization of austenite in the supersaturated dendrite, but not in the austenite in the eutectic phase.
Point 6: General: replace “dendrite austenite” by “primary (austenitic) dendrite”, “eutectic austenite” by “austenite in eutectic”
Response 6: Revised in the text (p.1, line.23-24, p.10, line.208, p.22, line.320, p.23, line.335, p.30, line. 460-462)
Point 7: Line 411: “it is very keen to chemical composition” à “it is very sensitive to chemical composition”
Response 7: Revised in the text (p.30, line. 464)
Point 8: Line 413: In which way relate the mechanical properties to the composition? Please be more specific.
Response 8: Revised in the text: ‘The mechanical properties such as hardness and wear resistance are closely related to the fractions of the existing phases.’ (Which resulted from the chemical composition such as modified carbon equivalent). (p.31, line. 466-467)

Reviewer 2 Report
Metals (ISSN 2075-4701)
metals-1396447
Effect of dendrite fraction on the M23C6 precipitation behavior and the mechanical properties of High Cr white irons
- Please include the method employed for determining chemical composition of the alloys.
- Authors informed: “Volume fraction of the primarily solidified dendrites in the as-cast specimens was measured during optical microscopy by Image analyzer”. How many images per condition? Which magnification?
3.Please inform thermal gradient adopted for the Directional Solidification and Quenching (DSQ) experiment.
- Figure 2 is hard to see the numbers and identification. Please increase the font size.
- It is well known that Cr is hard to fast diffuse into the Fe matrix as well as interfere in the C diffusion (decreasing C mobility). As such, equilibrium calculations are far from the reality. Please include some Scheil calculations (non-equilibrium rule) in order to discuss that.
- The authors afirmed: “It is expected to be caused by the steady state solidification characteristics during directional solidification.”. Please give referents for that and explain better this argument.
- Most of the profiles in Fig. 15 are quite constant. Are they really correlated with Ceq?
Author Response
Response to Reviewer 2 Comments
-Thank you for the kind and detailed comments!
Point 1: Please include the method employed for determining chemical composition of the alloys.
Response 1: To get varies dendrite fraction. (p.3, line.100-102)
Point 2: Authors informed: “Volume fraction of the primarily solidified dendrites in the as-cast specimens was measured during optical microscopy by Image analyzer”. How many images per condition? Which magnification?
Response 2: Addressed in the text. (p.5, line.122-123)
Point 3: Please inform thermal gradient adopted for the Directional Solidification and Quenching (DSQ) experiment.
Response 3: Addressed in the text: 100oC/cm at solid/liquid interface. (p.5, line.142-143)
Point 4: Figure 2 is hard to see the numbers and identification. Please increase the font size.
Response 4: Revised in Figure 2. (Changed Figure 2 to Figure1) (p.9, line.176-177)
Point 5: It is well known that Cr is hard to fast diffuse into the Fe matrix as well as interfere in the C diffusion (decreasing C mobility). As such, equilibrium calculations are far from the reality. Please include some Scheil calculations (non-equilibrium rule) in order to discuss that.
Response 5: It is right, however, Scheil calculation is another area of our main purpose. We will try to do that in the future study.
Point 6: The authors afirmed: “It is expected to be caused by the steady state solidification characteristics during directional solidification.”. Please give referents for that and explain better this argument.
Response 6: Revised the text and gave referent. (p.13, line.257-262)
Point 7: Most of the profiles in Fig. 15 are quite constant. Are they really correlated with Ceq?
Response 7: We gave variation of carbon equivalent in alloy design to get various fraction of primary dendrite. Thus, the mechanical properties of the alloys were plotted according to carbon equivalent. The values have much difference in the as-cast from those of the heat treated condition. The increase of the values was more in the alloys with high fraction of primary dendrite.

Reviewer 3 Report
p.1-2 Introduction
Except for the last two lines, it is difficult to find the relation with the present study. It looks like a kind of summary of textbooks. Please mention about the knowledge gaps and how it is approached in the present study. The last two lines are mentioned about the present study, but it is too simple. Please write more details (what has been done to fill the knowledge gaps).
p.2, lines 81–82, “2.1 ~2.9% 81 C, and 24.0 ~ 27.0% Cr.”
I believe that these are wt%. Please specify it to avoid confusion.
p.3, lines 85, “ThermoCalc”
Please write the version of ThermoCalc and database name/version.
p.3, Table 1
I believe that these are wt%. Please specify it to avoid confusion.
p.5, Table 2
It is written as “Phase fraction” and “Primarily solidified dendrite fraction (%)”, but these should be mole fraction (or mol %) according to the ThermoCalc calculation, which is shown in Figure 2. Please specify it to avoid confusion.
p.6, Figure 2
The resolution is too low. It is hard to see.
p.7, Lines 158–159, “The alloy with fully eutectic structure 2927 shows the phase fraction of austenite 158 to M7C3 is 2.76:1. Except”
At which temperature? According to Table 2, it seems at the final freezing temperature (1284C), but it is difficult to follow the discussion. Please add it in the text.
p.7, Table 3 and lines 186–189 (discussion on the difference between calc. and measured values).
I think the measured dendrite fraction is the volume fraction (or area fraction). On the other hand, the calculated value is the mole fraction. It is nonsense to compare the values directly.
All discussion for the comparison between the calculated and measured ”fraction” must be rewritten (e.g. section 3.4.2.1 at page 18).
p.24, Fig. 15
How were the curves between the data points drawn?
For instance, there is a local maximum at Ceq=3.6 and a local minimum at Ceq=3.9 for DS+WQ+Age. Is there any reason? If it is just a smooth curve, there is no meaning to draw it.
p.25, Fig. 16
How were the curves between the data points drawn? If it is just a smooth curve, there is no meaning to draw it.
Author Response
Response to Reviewer 3 Comments
-Thank you for the kind and detailed comments!
Point 1: p.1-2 Introduction
Except for the last two lines, it is difficult to find the relation with the present study. It looks like a kind of summary of textbooks. Please mention about the knowledge gaps and how it is approached in the present study. The last two lines are mentioned about the present study, but it is too simple. Please write more details (what has been done to fill the knowledge gaps).
Response 1: Revised the Introduction. (p.1-2, line.29-84)
Point 2: p.2, lines 81–82, “2.1 ~2.9% 81 C, and 24.0 ~ 27.0% Cr.”
I believe that these are wt%. Please specify it to avoid confusion.
Response 2: Addressed in the text. (p3, line.100-101)
Point 3: p.3, lines 85, “ThermoCalc”
Please write the version of ThermoCalc and database name/version.
Response 3: Addressed in the text. (p.3, line.105-106)
Point 4: p.3, Table 1
I believe that these are wt%. Please specify it to avoid confusion.
Response 4: Revised the text. (p,4, line.114-115)
Point 5: p.5, Table 2
It is written as “Phase fraction” and “Primarily solidified dendrite fraction (%)”, but these should be mole fraction (or mol %) according to the ThermoCalc calculation, which is shown in Figure 2. Please specify it to avoid confusion.
Response 5: Revised the text and table. (Changed Figure 2 to Figure 1) (p.7, line.171)
Point 6: p.6, Figure 2
The resolution is too low. It is hard to see.
Response 6: Enlarged the figure to get clear characters. (Changed Figure 2 to Figure 1) (p.9, line.176-177)
Point 7: p.7, Lines 158–159, “The alloy with fully eutectic structure 2927 shows the phase fraction of austenite 158 to M7C3 is 2.76:1. Except”
At which temperature? According to Table 2, it seems at the final freezing temperature (1284C), but it is difficult to follow the discussion. Please add it in the text.
Response 7: Added in the text and deleted “Exept~also.”. (p.10, line.185-186, 194)
Point 8: p.7, Table 3 and lines 186–189 (discussion on the difference between calc. and measured values).
I think the measured dendrite fraction is the volume fraction (or area fraction). On the other hand, the calculated value is the mole fraction. It is nonsense to compare the values directly.
All discussion for the comparison between the calculated and measured ”fraction” must be rewritten (e.g. section 3.4.2.1 at page 18).
Response 8: Rewrote in the text.: It was possible to find the propensity of the phase evolution during solidification. (p.10, line.206-218)
Point 9: p.24, Fig. 15
How were the curves between the data points drawn?
For instance, there is a local maximum at Ceq=3.6 and a local minimum at Ceq=3.9 for DS+WQ+Age. Is there any reason? If it is just a smooth curve, there is no meaning to draw it.
Response 9: We tried to find the relationship the increase of hardness and primary dendrite fraction (or Carbon equivalent) Also, we tried to find the effect of heat treatment. We substituted Table data to the Figure. (p.28, line.423-424, Table 5.)
Point 10: p.25, Fig. 16
How were the curves between the data points drawn? If it is just a smooth curve, there is no meaning to draw it.
Response 10: The Figure was switched to Table data. (p.29, line.444-445, Table 6.)

Round 2
Reviewer 3 Report
The manuscript has been revised based on the reviewers' comments and now it is acceptable for the publication.